# Abundance of Phasi-Charoen-like virus in *Aedes aegypti* mosquito populations in different states of India

**Kavita Lole, Ashwini Ramdasi, Sucheta Patil⦿, Shivani Thakar, Amol Nath, Onkar Ghuge, Abhranil Gangopadhayya, Anakkathil B. Sudeep\*, Sarah Cherian\***

ICMR- National Institute of Virology, Pune, Maharashtra, India

\* sudeepmcc@yahoo.co.in (ABS); sarahcherian100@gmail.com (SC)

## Abstract

Mosquitoes are known to harbor a large number of insect specific viruses (ISV) in addition to viruses of public health importance. These ISVs are highly species specific and are non-pathogenic to humans or domestic animals. However, there is a potential threat of these ISVs evolving into human pathogens by genome alterations. Some ISVs are known to modulate replication of pathogenic viruses by altering the susceptibility of vector mosquitoes to pathogenic viruses, thereby either inhibiting or enhancing transmission of the latter. In the present study, we report predominance of Phasi Charoen-like virus (PCLV, Family: *Phenuviridae*) contributing to >60% of the total reads in *Aedes aegypti* mosquitoes collected from Pune district of Maharashtra state using next generation sequencing based metagenomic analysis of viromes. Similar results were also obtained with mosquitoes from Assam, Tamil Nadu and Karnataka states of India. Comparison of Pune mosquito sequences with PCLV Rio (Brazil) isolate showed 98.90%, 99.027% and 98.88% homologies in the S, M and L segments respectively indicating less genetic heterogeneity of PCLV. The study also demonstrated occurrence of transovarial transmission as seen by detection of PCLV in eggs, larvae, pupae and male mosquitoes. *Ae. aegypti* mosquitoes collected from Pune also showed a large number of reads for viruses belonging to *Baculoviridae*, *Rhabdoviridae*, *Genomoviridae* and *Bunyaviridae* families. The role of PCLV in the replication of dengue and chikungunya virus is yet not clear. It warrants further studies to know the significance of PCLV and other ISVs on the replication and transmission of *Ae. aegypti* borne pathogenic viruses, especially in the absence of prophylactics or therapeutics.

## Introduction

Emerging and re-emerging arthropod borne viruses continue to be a major threat to humanity as they not only inflict high morbidity and mortality but also devastate the economy of many countries especially in the tropical and subtropical countries [1–4]. During the last few decades, mosquito-borne viruses have emerged as a major public health concern, replacing malaria that had the dubious distinction as a major mosquito borne killer disease across the

**Data Availability Statement:** All relevant data are within the article and its supporting information files.

**Funding:** Indian Council of Medical Research, New Delhi, India funded the project as an extramural project granted to ABS, KSL and SC. Grant No. 6/9-7 (2019-ECD-II). The funders had no role in study design, data collection and analyis, decision to publish or preparation of the manuscript.

**Competing interests:** The authors have declared that no competing interest exist.

globe. Recent efforts on diagnosis and therapy have brought down malaria cases in most of the endemic countries, while therapeutics/vaccines for arboviral infections still remain scanty. The increased frequency and speed of human travel and commercial trade combined with global warming has resulted in geographic expansion of mosquito vectors and their pathogens from indigenous habitats to newer/naïve areas. The emerging /re-emerging viruses have the potential of inducing large fatal outbreaks involving the whole country or a continent in a short time span. The chikungunya virus (CHIKV) which re-emerged in Kenya in 2004, spread like a wild fire causing large-scale outbreaks with high morbidity in several countries of Asia, Europe, Oceania, the Caribbeans and the South and North Americas [1,2]. Similar is the case with West Nile virus (WNV), that caused massive outbreaks in North America, South America and Europe with high morbidity and mortality, since it was first detected in New York in 1999 [3,4].

*Aedes* mosquitoes, especially *Aedes aegypti* has garnered special attention due to its rapid geographical spread and public health importance. It has already established itself in the tropical and subtropical climate zones. The day biting urban mosquito is highly anthropophilic and is the principal vector of some of the important viruses, *viz*., dengue, chikungunya, yellow fever and Zika [5]. Dengue is endemic to almost all the tropical and subtropical countries with estimated 390 million cases annually [6]. In the absence of effective vaccines or therapeutics as well as mosquito control, dengue continues to be a major health concern today. The control of arboviral infections still largely relies on the vector management. Recent studies documenting reduced replication and transmission of dengue virus in *Aedes* mosquitoes infested with bacteria of *Wolbachia* species seems to be a promising novel biological control means [7,8]. Promising results have also been observed with certain insect specific viruses (ISVs) that alter the susceptibility of vector mosquitoes or cell lines derived from mosquitoes to human pathogenic viruses [9,10]. These studies have shown inhibition/ reduced replication of Japanese encephalitis, Murray Valley encephalitis and West Nile viruses in respective vector mosquitoes/ mosquito cells that were experimentally infected with ISVs. These studies suggest possibility of using ISVs as biocontrol agents since they cause homologous interference in the hosts by blocking the receptors [11]. However, ISVs are also regarded as the precursor to several human pathogenic viruses and therefore need special attention as they have the potential to emerge and cause human infections [12].

Studies conducted in China, Grenada and other countries have reported the prevalence of Phasi Charoen-like virus (PCLV) in *Aedes aegypti* populations, however, its potential to influence replication of viruses like dengue and chikungunya is not yet clear [13,14]. In the present study, we report predominance of PCLV in *Aedes aegypti* mosquitoes collected from Pune, Maharashtra, and three other states of India, *i.e*., Karnataka, Tamil Nadu and Assam.

## Materials and methods

### (i) Mosquito collection and processing

Adult *Aedes aegypti* mosquitoes were collected using hand held mouth aspirators, identified using entomological keys [15], pooled according to gender and locality and stored at -86°C until processing. *Aedes* larvae collected from breeding habitats were brought to ICMR-National Institute of Virology (NIV), Pune laboratory and reared until adults. The larvae reared adults were identified individually, pooled and used for the study. The mosquitoes were maintained at 28°C with 80–85% humidity and a 12:12 h photoperiod. Mosquitoes from Pune district of Maharashtra comprised collections from five different locations, *viz*., Dange Chowk, Savitribai Phule Pune University campus, Tadiwala Road, Alandi and Chincholi while that of Karnataka and Assam represented by mosquitoes from Bengaluru and Dibrugarh respectively.

The Tamil Nadu mosquitoes represented the laboratory colony established at NIV from mosquito eggs received from ICMR-Vector Control Research Centre, Madurai, Tamil Nadu (Fig 1). Only female mosquitoes obtained from five locations of Pune were processed for the total virome analysis. For other locations, adult mosquitoes, eggs, larvae and pupae were processed for RT-PCR based detection of PCLV using S region specific primers (Table 1).

## (ii) Next generation sequencing based *Aedes aegypti* virome analysis using Oxford Nanopore technology

**Mosquito homogenization and sample processing.** Viral RNA was extracted from pools of 20–50 *Aedes aegypti* female mosquitoes. Briefly, the mosquitoes were surface sterilized using absolute ethanol for first wash followed by two consecutive washes with 70% ethanol and a final wash with sterile milliQ water/nuclease-free water. Surface sterilized mosquitoes were transferred to a sterile pre-chilled mortar and the frozen mosquito tissue was crushed in liquid nitrogen using a pestle until the mosquito tissue was completely homogenized as a fine powder. The homogenized tissue was suspended in 1X PBS (phosphate buffered saline) containing 5 mM $MgCl_2$ and 1.4 mM Dithiothreitol (DTT, Invitrogen) and centrifuged at 17,000 g for 10 min at 4˚C to pellet down cellular debris.

**Viral RNA enrichment.** Clear supernatants were collected and filtered twice through sterile 0.22 μM syringe filters to exclude bacterial and other larger microscopic entities. Filtrates were treated with 2 U/μl TURBO DNase (Invitrogen), 10 U/μl RNase A (Invitrogen) and incubated at 37˚C for one hour to digest mosquito nucleic acids, bacterial and other genomes, essentially to enrich the enveloped/capsid-protected viruses.

## Optimization of the workflow for NGS based virome analysis

To optimize the workflow for library preparation, lysates were prepared using pools of 50 *Ae. aegypti* female mosquitoes (from laboratory colony) and spiked with cell culture grown West Nile virus (WNV), Japanese encephalitis virus (JEV), dengue virus (DENV) (serotype two),

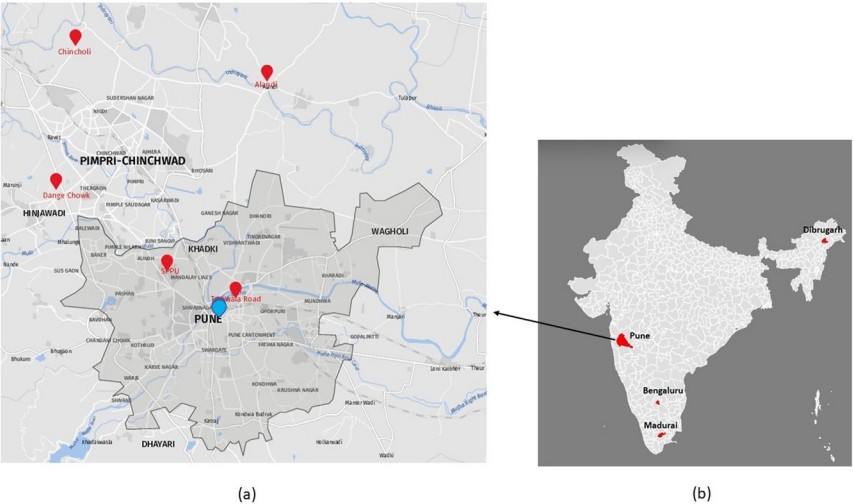

(a)                                    (b)

**Fig 1. Geo-reference map of locations from where the mosquitoes were collected.** 1a) The enlarged map of Pune Municipal Corporation (PMC), Pune City, prepared using software available at https://www.mapz.com (with due permission) shows two locations (Savitribai Phule Pune University area and Tadiwala road) while three locations on the outskirt of Pune city (Alandi, Chincholi and Dange chowk), 1b) Map of India prepared using software at https://www.gramener.com (free software) showing four locations from India, *viz*., Bengaluru, Dibrugarh, Madurai and Pune.

**Table 1. Primers used in the study.**

| Primer type | Primer | Primer sequence (5' to 3') | Reference |
|---|---|---|---|
| Random anchored primer | RT5 | CATCACATAGGCGTCCGCTGNNNNNNN | Xiao et al. [16] |
| Random anchored primer | RT10 | CGACCCTCTTATCGTGACGGNNNNNNN | Xiao et al. [16] |
| Random anchored primer | RT12 | GGTGGGCGTGTGAAATCGACNNNNNNN | Xiao et al. [16] |
| Random anchored primer | IDT- K-8N | GACCATCTAGCGACCTCCACNNNNNNNNN | Chrzastek et al. [17] |
| Anchor primer | P5 | CATCACATAGGCGTCCGCTG | Xiao et al. [16] |
| Anchor primer | P10 | CGACCCTCTTATCGTGACGG | Xiao et al. [16] |
| Anchor primer | P12 | GGTGGGCGTGTGAAATCGAC | Xiao et al. [16] |
| Anchor primer | IDT-K | GACCATCTAGCGACCTCCAC | Chrzastek et al. [17] |
| L segment- Forward | | AGACAGCACAAGCAAATAAAGCAAG | Ramos-Nino et al. [13] |
| L segment-Reverse | | AAACATGCATTGTAAGGTTTTGTCG | Ramos-Nino et al. [13] |
| M segment-Forward | | AAAAGTAGGAATTGATGCTGTTGC | Ramos-Nino et al. [13] |
| M segment-Reverse | | CTTTGAGCACTTTTGTCTAATGGC | Ramos-Nino et al. [13] |
| S segment Forward | | CAGTTAAAGCATTTAATCGTATGATAA | Ramos-Nino et al. [13] |
| S segment Reverse | | TGGAAAATAAAAACAATAAAGCAATAC | Ramos-Nino et al. [13] |

Zika virus (ZIKV) and chikungunya virus (CHIKV) particles, equivalent to $10^4$ plaque forming units each (to mimic realistic viral loads in mosquitoes) and processed further for library preparation. The primers for cDNA synthesis and amplification were selected based on broad metagenomic sensitivity for pathogen detection from previous reports [16–20]. A two-step strategy was optimized for cDNA synthesis using random anchored primers wherein first strand synthesis was done using reverse transcriptase while the second strand synthesis was done by using Klenow fragment. Four, out of fourteen, random anchored primers were selected for carrying out cDNA synthesis based on the properties, *i.e.*, low self-annealing tendency, similar annealing temperatures and low tendency of primer dimer formation. Primer concentrations, reaction conditions and thermal cycling conditions were optimized to obtain ds-cDNA concentration equivalent to the input viral RNA template concentration. Sequence Independent Single Primer Amplification (SISPA) strategy was used for amplification of the cDNA fragments generated from viral RNA genomes.Library preparation was carried out using standard protocols provided by the Oxford Nanopore Technologies (ONT) for the Nanopore platform as described below using different primer sets (Table 1). The reaction conditions that yielded maximum sequence coverage (>90%) and depth for the spiked virus genomes were further used for processing the field collected mosquitoes from Pune.

## Viral RNA extraction and library preparation

Viral RNA was extracted using QIAamp Viral RNA Mini Kit (Qiagen) as per the manufacturer's instructions. Additional two washes with 80% and 70% ethanol were given on the column before sample elution to remove PCR inhibitors and to improve the quality of RNA. The eluted RNA was quantified using Qubit flourometer (Invitrogen).

## Reverse transcription and double strand cDNA synthesis

RNA was subjected to first strand cDNA synthesis using four random anchored primers at a concentration of 20 picomoles (Table 1). For that, random anchored primers; one primer to each reaction, were added and incubated at 75˚C for 5 min to allow denaturation of RNA secondary structures; and subjected to snap-cooling by placing on ice for 5 min to allow annealing of the primers to the template RNA. The first strand reaction final volume of 20 μl consisted of

input RNA and primer, 1.0 µl SuperScript III Reverse Transcriptase (RT) (Invitrogen), 1.0 µl 0.1M DTT (Invitrogen), 1.0 µl (40U) RNase OUT (Invitrogen), 1.0 µl 25mM dNTPs and 4.0 µl 5 X first-strand buffer (Invitrogen). The reaction was carried out as follows: 25˚C 10 min, 50˚C 60 min, and 75˚C 10 min. The first-strand cDNA synthesis reaction was followed by treatment with 250 U RNase H (NEB; New England Biologicals) to digest template RNA strand. Second strand cDNA synthesis was carried out using Klenow fragment (NEB). For that, the corresponding random anchored primer (Table 1) was added to the first-strand reaction and incubated at 75˚C for 5 min, followed by snap-cooling on ice for 5 min. The final volume of 30 µl consisted of the first strand reaction, 10 picomoles of random anchored primer, 2.0 µl 10X Klenow buffer (NEB) and 200 U of Klenow fragment (NEB). The reaction was carried out at 37˚C for 60 min followed by heat inactivation at 70˚C for 10 min.

## Sequence-Independent Single-Primer Amplification (SISPA)

The double stranded cDNA was subjected to SISPA, using anchor primers corresponding to the respective random anchored primers (Table 1). The 50 µl SISPA reaction consisted of 5.0 µl ds-cDNA, 2.5µl anchor primer, 1.0 µl 25mM dNTPs, 0.5µl AmpliTaq Polymerase (Invitrogen), 10X PCR buffer (Invitrogen) and 1.25 µl 25mM $MgCl_2$ (Invitrogen). The reaction was carried out at 95˚C for 3 min, followed by 35 cycles at 95˚C for 20 sec; 54˚C for 45 sec; 72˚C for 1 min and a final extension at 72˚C for 7 min.

SISPA reaction was followed by a magnetic bead purification step using AMPure XP magnetic beads (Beckman-Coulter) to purify amplicons as per the manufacturer's instructions. The amplicons were eluted in nuclease-free water and quantified using Qubit fluorometer. Only samples with quantities higher than 100 µg were taken forward for library preparation to ensure inclusion and representation of viral species from mosquito samples.

## Library preparation and sequencing on Oxford Nanopore MinION Mk1B sequencer

DNA repair, blunt-end preparation and adapter ligation of amplicons were done as per the standard protocol for library preparation recommended by ONT using NEBNext FFPE DNA Repair Mix (M6630) and NEBNext Ultra II End repair / dA-tailing Module (E7546). The DNA sample was then subjected to clean-up with AMPure XP beads, eluted and quantified on Qubit fluorometer. Approx. 200–300 ng of amplified DNA of each sample (100 to 200 femtomoles) was ligated to sequencing adapters using SQK-LSK 109 kit (ONT) and barcoding was done using native barcoding EXP-NBD104 kit (ONT). Equal quantities of barcoded samples were pooled to 300ng for final sequencing and the remaining libraries were stored at -80˚C. Adapter ligated libraries were loaded onto the SpotON flow-cell (R.9.0.4) (MinION Mk1B system) and sequencing runs were performed for about 24 h using MinKNOW software (ONT).

**Basecalling and data QC.**   The generated fast5 files were processed for basecalling using Guppy_basecaller v4.0.15 with high accuracy config file (dna_r9.4.1_450bps_hac.cfg). The reads in fastq format were demultiplexed (only if both the ends had the barcodes) using Guppy_barcoder v4.0.15. Only the reads with a minimum q-score of 7 were considered for further analysis. Porechop v0.2.4 was used for trimming all the adapter and barcode sequences. The resulting clean reads were used for further analysis.

**Metagenomic classification.**   The reads were classified using Centrifuge v1.0.4 against a database of viral genomes. The genome specific reads were sorted and mapped against respective genomes using minimap2 and the coverage was calculated. The bcftools was used for assembling the mapped reads and generating the consensus genome sequence.

## Confirmation of presence of PCLV with conventional RT-PCR and sequencing

Fresh viral RNA was extracted from Pune mosquito pools as described above. Forward and reverse pairs of primers as given in Table 1 were used for amplification of ~900–1000 nucleotide sequences from L, M and S segments of PCLV genome by RT-PCR using SuperScript™ III One-Step RT-PCR System with Platinum™ Taq DNA Polymerase (Invitrogen). The amplicons were purified and both strands were sequenced using BigDye Terminator v3.1 Cycle Sequencing Kit (Thermo Fisher Scientific) and analyzed on ABI 3130xl Genetic analyzer (Applied Biosystems, Foster City, USA). Mosquito samples collected from different states were processed similarly for viral RNA isolation and used for S gene amplification and sequencing.

## Vertical transmission of PCLV

Eggs, larvae, pupae and adult mosquitoes reared from *Ae. aegypti* eggs from Madurai (Tamil Nadu) were processed to determine whether vertical transmission of the virus is occurring in the *Aedes aegypti* mosquitoes of the region. Generated amplicons were processed for sequencing and phylogenetic analysis.

## Phylogenetic analysis

The obtained sequences were trimmed to remove primer sequences and aligned with various reference sequences downloaded from NCBI GenBank, using the ClustalW algorithm available through the Molecular Evolutionary Genetics Analysis (MEGA) 5 software [21]. Following this, the best fitting model for phylogenetic analysis was computed, and found to be the Tamura 3-Parameter model [22]. Maximum Likelihood phylogenetic trees were constructed using this model, taking uniform rates and 1000 bootstrap replicates.

# Results

## Optimization of library preparation for metagenome analysis

Lysates prepared from the laboratory colony of mosquitoes were spiked with cell culture grown WNV, JEV, dengue (serotype two), ZIKV and CHIKV particles and processed for viral RNA isolation, cDNA synthesis and library preparation using four different sets of primers. This was done in order to have the best conditions to carry out the metagenomic virome analysis of field mosquitoes. Library prepared with each of the P5 and P10, P12 and IDT-K primer set generated ~3 million reads. The trimmed reads were aligned onto the reference genomes of dengue, CHIKV, WNV, ZIKV, JEV and *Aedes aegypti* genome using minimap2, independently. All four primers, P5 and P10, P12 and IDT-K amplified the spiked viruses generating reads with the N50 of around 698 bases. Among the spiked viruses, dengue genome was recovered with maximum coverage 99.5%, while WNV, JEV and ZIKV genomes showed ~90% coverage with P5 and P10 primers. There were no reads representing chikungunya virus with both P5 and P10 primers. Though both P12 and IDT-K primers were able to successfully amplify CHIKV sequences, IDT-K primer (random octamer) showed >90% genome recovery. Pooling of libraries generated with primers P10 and IDK-K yielded sequence reads showing >90% coverage of genomes of all spiked viruses and hence it was decided to use both P10 and IDT-K primers for amplification in further experiments of virome analysis.

## Metagenomic virome analysis of field caught *Aedes aegypti* mosquitoes

Field caught mosquitoes were processed for library preparation and sequencing using protocols optimized with virus spiking experiment. Each total virome sequencing run generated ~6.0 Gb data after 24 hours of the sequencing. After basecalling, demultiplexing, and trimming of adapters and primer sequences, average length of the reads was found to be ~300 bases (S1 Table). After removing the host *Ae. aegypti* sequences and reads <100 of length, the remaining clean reads were classified using BLASTn tool. The percentage of classified reads ranged between 3 to 53% for different runs after classification against a database containing human and viral genomes from RefSeq database (Table 2).

The virus metagenomic analysis revealed abundance of Phasi Charoen-like virus (PCLV) belonging to *Phenuiviridae* family of the order *Bunyavirales* in all the mosquito samples from Pune. PCLV constituted an average of 62.25% of the total mapped viral reads. On further mapping of these against the PCLV L, M and S genome segments to estimate the genome coverage, eleven out of seventeen processed mosquito pool samples showed more than 70% coverage of PCLV genome, while two samples showed more than 90% virus genome coverage (S2 Table). The metagenome sequences generated from *Ae. aegypti* mosquitoes from Pune are deposited in GenBank (S3 Table; Accession Nos. five locations and SRA IDs respectively). In addition to PCLV, reads of other viruses, *viz., Choristoneura fumiferana* granulovirus (ChfuGV) (*Baculoviridae*), Piry vesiculovirus (PIRYV) (*Rhabdoviridae*), Human Gemykibivirus 2 (HuGkV-2) (*Genomoviridae*), Shamonda virus (*Bunyaviridae*) were also detected in the samples (Table 3). Since the reads for these viruses were comparatively few and genome coverage and depth were comparatively low, these sequences were not analyzed further.

**Table 2. Classified reads after classification against a database containing human and viral genomes.**

| Sample | Classified reads | % classified reads | Chordate reads | Viral reads | PCLV reads | PCLV % in virome |
|---|---|---|---|---|---|---|
| RUN2_barcode01 | 18573 | 6.93 | 8343 | 10209 | 9685 | 94.87 |
| RUN2_barcode03 | 6408 | 2.88 | 3529 | 2877 | 2804 | 97.46 |
| RUN2_barcode04 | 522282 | 51.29 | 521008 | 1259 | 604 | 47.97 |
| RUN2_barcode05 | 141041 | 30.34 | 131983 | 8901 | 3013 | 33.85 |
| RUN2_barcode06 | 255523 | 30.02 | 242491 | 12906 | 8927 | 69.17 |
| RUN2_barcode07 | 322213 | 48.38 | 287419 | 34736 | 29867 | 85.98 |
| RUN2_barcode08 | 87601 | 14.72 | 47884 | 39541 | 33189 | 83.94 |
| RUN3_barcode01 | 340 | 38.12 | 336 | 4 | 3 | 75.00 |
| RUN3_barcode02 | 1021 | 39.93 | 1005 | 16 | 12 | 75.00 |
| RUN3_barcode03 | 1825 | 51.77 | 1755 | 70 | 22 | 31.43 |
| RUN3_barcode04 | 528 | 13.14 | 341 | 182 | 105 | 57.69 |
| RUN3_barcode05 | 258 | 12.60 | 179 | 79 | 44 | 55.70 |
| RUN4_barcode01 | 201193 | 11.72 | 121277 | 79145 | 44009 | 55.61 |
| RUN4_barcode02 | 154833 | 15.34 | 125205 | 29520 | 21630 | 73.27 |
| RUN4_barcode03 | 260004 | 34.92 | 257119 | 2825 | 1934 | 68.46 |
| RUN4_barcode04 | 132896 | 14.65 | 122883 | 9914 | 7331 | 73.95 |
| RUN4_barcode05 | 362075 | 29.01 | 354058 | 7826 | 2923 | 37.35 |
| RUN5_barcode01 | 325579 | 52.86 | 323486 | 2031 | 734 | 36.14 |
| RUN5_barcode02 | 91354 | 34.45 | 89156 | 2147 | 1059 | 49.32 |
| RUN5_barcode03 | 90009 | 28.16 | 83713 | 5674 | 2432 | 42.86 |
| RUN5_barcode04 | 20988 | 15.63 | 20272 | 696 | 9 | 1.29 |
| RUN5_barcode05 | 846 | 25.29 | 826 | 20 | 8 | 40.00 |
| RUN5_barcode06 | 1390 | 13.78 | 1205 | 184 | 8 | 4.35 |

**Table 3. Species identified with minimum 50 reads.**

| Species | Taxon ID | Maximum reads classified | RUN2_NB01 | RUN2_NB03 | RUN2_NB04 | RUN2_NB05 | RUN2_NB06 | RUN2_NB07 | RUN2_NB08 | RUN4_NB01 | RUN4_NB02 | RUN4_NB03 |
|---|---|---|---|---|---|---|---|---|---|---|---|---|
| Homo sapiens | 9606 | 521008 | 8343 | 3529 | 521008 | 131983 | 242491 | 287419 | 47884 | 121277 | 125205 | 257119 |
| Phasi Charoen-like phasivirus | 1980610 | 44009 | 9685 | 8804 | 6604 | 3013 | 8927 | 29867 | 33189 | 44009 | 21630 | 1934 |
| Choristoneura fumiferana granulovirus | 56947 | 21118 | 166 | 12 | 14 | 188 | 203 | 2239 | 2620 | 21118 | 5774 | 296 |
| Piry vesiculovirus | 1972575 | 12405 | 209 | 7 | 569 | 5519 | 3622 | 2440 | 2057 | 12405 | 1475 | 512 |
| Human associated gemykibivirus 2 | 2004957 | 1011 | 1 | 0 | 2 | 77 | 1 | 82 | 1011 | | | |
| Shamonda orthobunyavirus | 159150 | 727 | 4 | 0 | | 5 | 3 | 11 | 38 | 174 | 47 | 48 |
| Cotesia congregata bracovirus | 39640 | 112 | 0 | 9 | | 15 | | | 91 | 112 | | |
| Lactobacillus virus LP65 | 298338 | 73 | 13 | 0 | 1 | 2 | 1 | 7 | | 13 | 18 | 2 |
| Prochlorococcus phage P-SSM2 | 268746 | 73 | 0 | 0 | | | | | | 10 | | |
| Mycobacterium virus Goose | 1211282 | 57 | 0 | 0 | | | | | 57 | | | |
| Gokushovirinae Bog1183_53 | 1655646 | 54 | 0 | 0 | | | | | | | | |

## Confirmation of the presence of PCLV in *Aedes aegypti* mosquitoes by conventional RT-PCR and sequence analysis

Viral RNA isolated from *Ae. aegypti* female mosquitoes collected from one location in Pune was processed for amplification of partial L, M, and complete S segment of PCLV genome to validate the NGS results. All the three segments could be successfully amplified (Figs 2A and S1). Sequencing of amplicons obtained from these segments using Sanger's method confirmed these as PCLV sequences and validated the results obtained with NGS based metagenomic analysis. Sequence comparisons with the reference PCLV Rio isolate showed 98.90%, 99.027% and 98.88% homologies in the S, M and L segments respectively. These findings confirmed findings of NGS based metagenome analysis of viruses suggesting that PCLV is present abundantly in wild, field caught *Ae. aegypti* mosquitoes from Pune district, Maharashtra.

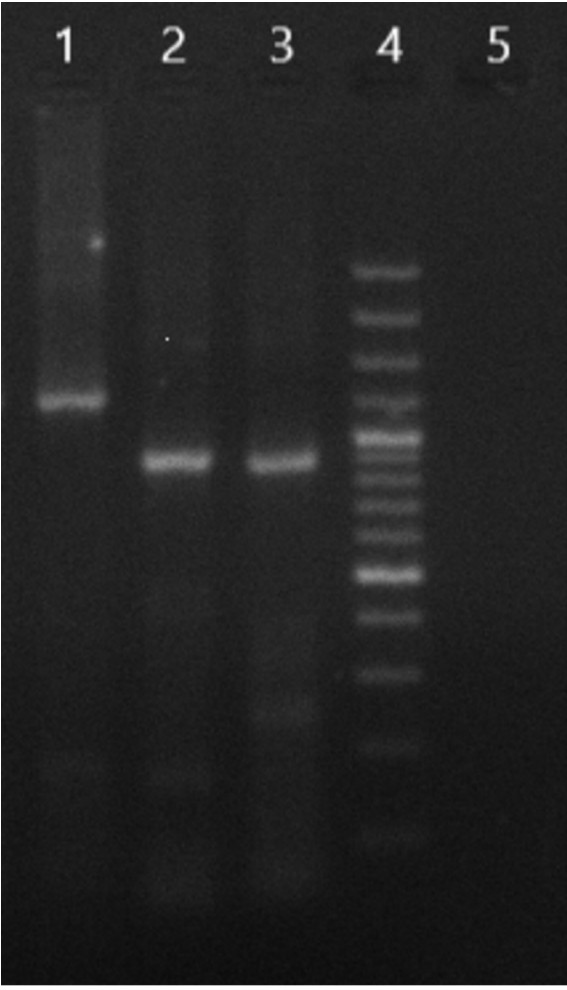

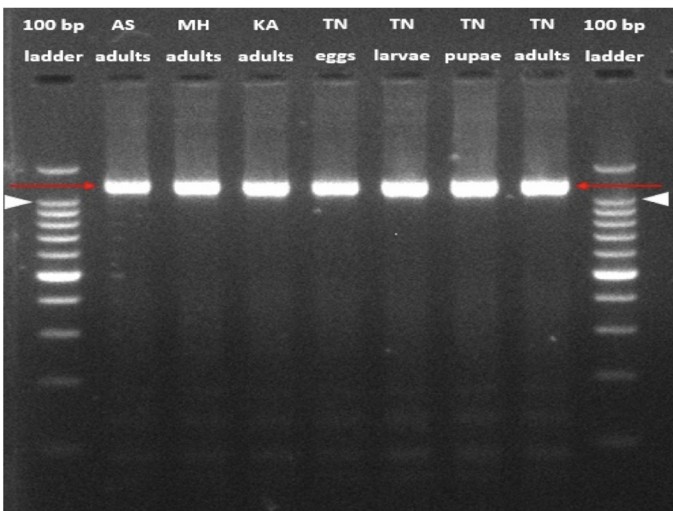

**Fig 2.  a: Detection of PCLV S, M and L segments from Pune adult female mosquito pools using RT-PCR.** Products were analysed on 2% agarose gel, lanes- L1: Complete S segment, L2: Partial M segment, L3: Partial L segment, L4: 100 bp DNA ladder, L5: Negative control. **b: Detection of PCLV S segment from mosquito samples from different locations from India using RT-PCR.** Products were analyzed on 2% agarose gel, lanes- L1: 100 bp DNA ladder (Invitrogen), L2: Assam (AS) adult mosquito pool, L3: Pune, Maharashtra (MH) adult mosquito pool, L4: Karnataka (KA) adult mosquito pool, L5: Tamil Nadu (TN) mosquito eggs, L6: Tamil Nadu (TN) mosquito larvae, L7: Tamil Nadu (TN) mosquito pupae, L8: Tamil Nadu (TN) mosquito adults, L9: 100 bp DNA ladder.

## Detection of PCLV in *Aedes aegypti* mosquitoes collected from other states of India

PCLV 'S' segment was amplified from *Ae. aegypti* mosquito samples collected from Karnataka, Assam and Tamil Nadu states of India (Figs 2B and S2). Presence of PCLV in *Ae. aegypti* mosquitoes in Karnataka has already been reported recently [23]. PCLV S segment sequences from different locations of India were compared to reference PCLV S segment sequences across the world. The lowest percent nucleotide identity observed was 99.90% between S segment sequences generated from *Ae. aegypti* mosquitoes from Assam, India, and adult mosquitoes from Rio (Brazil). The highest homology observed was 99.98% between sequences from pupae of Tamil Nadu mosquitoes and adults from Kenya (Table 4). *Ae. aegypti* mosquitoes from the four states showed >99.95% homology of PCLV 'S' gene sequences. Overall results indicate that PCLV is a highly conserved virus and viral strains circulating in *Ae. aegypti* mosquitoes across the world have very high genetic similarity suggesting a common ancestor for these strains (Fig 2B).

## Transovrial transmission of PCLV in *Aedes aegypti*

To determine whether transovarial transmission of PCLV occurs in *Ae. aegypti* mosquitoes, a colony was established from *Ae. aegypti* eggs obtained from Madurai (Tamil Nadu state) and larvae, pupae, adults and eggs from the colony were processed for S gene amplification. PCLV was detected in all the stages of *Ae. aegypti* indicating occurrence of transovarial transmission (vertical transmission) of the virus in *Ae. aegypti*. Nucleotide identities as compared to reference sequence were: eggs 99.984%, larvae 99.981, pupae 99.983% and adults 99.983% (Fig 3).

## Discussion

Metagenomic analysis using Next generation sequencing platforms has become a powerful tool for analysis of genomes of organisms. A study by Shi et al [24] has reported presence of >1500 novel RNA viruses in invertebrates. Subsequently, several investigators across the globe have used the tool and identified novel viral reads in important disease vectors *viz.*, *Aedes aegypti*, *Culex quinquefasciatus*, etc. The surge in dengue across the world in the recent decades has stimulated renewed research interest in *Ae. aegypti* mosquitoes as control of dengue still depends on vector management. It is speculated that vectorial capacity of mosquitoes can be influenced by the presence of endosymbiotic microbes by altering the susceptibility of the

**Table 4. Percent Nucleotide identities of PCLV S segment sequences from various life cycle stages of *Ae. aegypti* mosquitoes collected from different parts of India compared to PCLV sequences across the world.**

| Origin | NC_038263.1 Rio_Brazil | LC498493.1 Ghana | MH237597.1 Australia | MN053771.1 Guadeloupe | MT361067.1 Kenya | KM001087.1 Thailand | MN109951.1 Grenada | KU936055.1 UK | MH310081.1 USA | MF614134.1 China |
|---|---|---|---|---|---|---|---|---|---|---|
| **IND (TN) (eggs)** | 98.939 | 99.961 | 99.957 | 99.975 | 99.983 | 99.981 | 99.970 | 99.959 | 99.959 | 99.959 |
| **IND (TN) (larvae)** | 98.942 | 99.952 | 99.952 | 99.966 | 99.974 | 99.972 | 99.961 | 99.954 | 99.954 | 99.954 |
| **IND (TN) (pupae)** | 98.958 | 99.961 | 99.957 | 99.979 | 99.986 | 99.981 | 99.981 | 99.959 | 99.959 | 99.963 |
| **IND (TN) (female)** | 98.934 | 99.961 | 99.956 | 99.979 | 99.983 | 99.977 | 99.977 | 99.957 | 99.957 | 99.959 |
| **IND (MH)** | 98.901 | 99.950 | 99.941 | 99.956 | 99.967 | 99.959 | 99.952 | 99.943 | 99.943 | 99.946 |
| **IND (AS)** | 98.900 | 99.957 | 99.944 | 99.965 | 99.979 | 99.970 | 99.963 | 99.946 | 99.946 | 99.948 |
| **IND (KN)** | 98.901 | 99.948 | 99.939 | 99.957 | 99.968 | 99.961 | 99.954 | 99.941 | 99.941 | 99.941 |

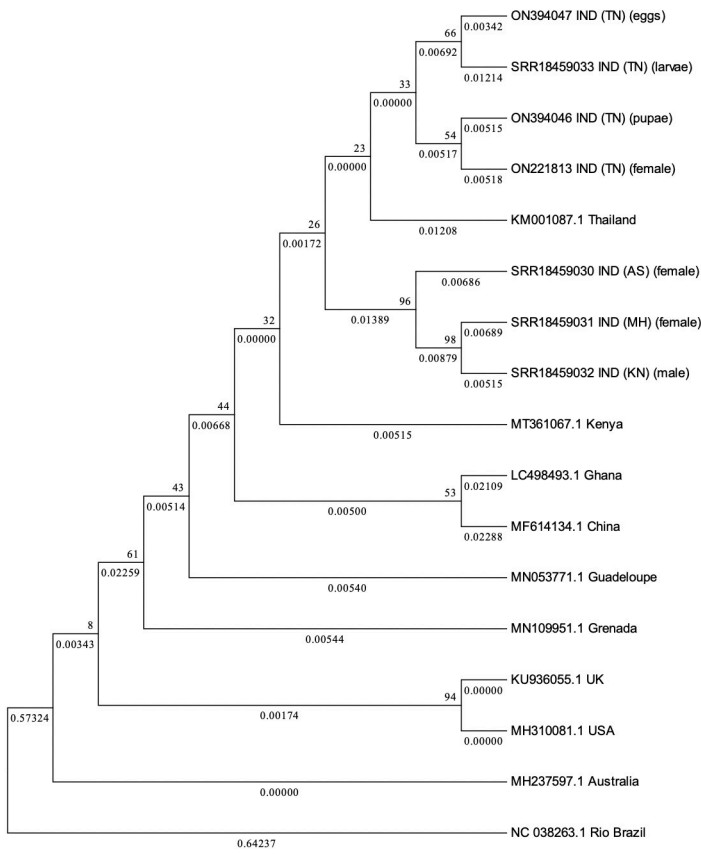

**Fig 3. Phylogenetic tree for PCLV S segment sequences.** Phylogenetic tree constructed via Maximum Likelihood (ML) method, using Tamura 3-Parameter and 1000 Bootstrap replicates for PCLV S segment sequences obtained from *Aedes aegypti* mosquitoes from different states of India, along with reference sequences from across the world.

mosquitoes to pathogenic viruses [25]. Studies have also shown that endosymbiont organisms in mosquitoes inhibit or reduce the replication of pathogenic viruses and could be used as biological control agents [10,13,24]. Studies using *Wolbachia*, an enosymbiont bacterium in mosquitoes, have also shown that it can inhibit the replication of many eukaryotic viruses [7,26]. *Aedes aegypti* mosquitoes, which lack the natural colonization of *Wolbachia* have shown resistance to dengue and chikungunya viruses when experimentally infected with *Wolbachia* [27]. *Culex quinquifasciatus* mosquitoes harboring *Wolbachia* have also shown resistance to West Nile virus upon infection [28]. Since no licensed vaccine is available for dengue today, these studies may evolve in the development of suitable control agents to limit the growth of the virus in *Aedes aegypti* mosquito.

In the present study, we analysed virome *of Ae. aegypti* mosquitoes collected from different parts of India, *i.e.*, Pune district of Maharashtra, Bengaluru (Karnataka), Dibrugarh (Assam) and Madurai (Tamil Nadu). We observed abundance of PCLV sequences in the mosquito samples tested from all the states. In complete virome analysis of field caught *Ae. aegypti* mosquitoes from Pune region of Maharashtra, PCLV constituted >60% reads of the total mapped viral reads. The rest constituted by an array of viruses, *viz*., members of virus families *Baculoviridae, Rhabdoviridae, Genomoviridae, Bunyaviridae* etc. Among these, Piry virus (PIRYV) (Family: *Rhabdoviridae*, genus: *Vesiculovirus*) and Human Gemykibivirus 2 (HuGkV-2) (family: Genomoviridae) (Table 3) are suspected to be human pathogens hence need further

attention [29]. Piry virus (PIRYV) (genus *Vesiculovirus*) is similar to Chandipura virus [30] that shows very high mortality in infected children. Serosurveys from different regions of Brazil have shown 4–17.7% seropositivity against PIRYV suggestive of human exposure to the virus. It was noted that accidental human exposure to PIRYV induces symptoms such as fever, headache, weakness, myalgia and arthralgia indicating that the virus has potential to cause human disease [31]. Importantly, experimental PIRYV infections in adult Swiss albino mice induces neuropathological damage and behavioral changes [32] suggesting severe manifestations upon infection. Hence PIRYV warrants further studies to assess potential threat to humans. Though clinical impact of HuGkV-2 is currently not known, detection of HuGkV-2 DNA in human blood samples from Brazilian Amazon also draws attention towards this virus [33]. Reads detected for another virus requiring attention was, Shamonda virus belonging to the Simbu group of the genus *Orthobunyavirus* of the family *Peribunyaviridae*. Orthobunyaviruses are transmitted between mammalian hosts by arthropod vectors. Their tripartite genomes facilitate reassortment of genomic segments. Detection of Shamonda virus in *Ae aegypti* adult mosquitoes collected from different locations from Pune indicates that these mosquitoes are persistently infected with the virus. PCLV also belongs to the same family. With rapid global movement and favorable climatic changes for emergence of new viruses, it would be essential to monitor Peribunyavirus activity in mosquitoes. It would be worthwhile to isolate these mosquito viruses and do further studies to understand whether these viruses have ability to interfere in the growth of each other and of the viruses of medical importance. The metagenomic virome analysis did not show presence of any viruses of medical importance such as DENV, CHIKV and ZIKV in the field caught mosquitoes, possibly since there were no current outbreaks. We used RNA metagenomic approach using the Oxford Nanopore sequencing platform to generate nearly complete genome sequence of Phasi Charoen-like virus infecting field-caught *Ae. aegypti* female mosquitoes in Pune, Maharashtra, India. Analysis of *Ae. aegypti* mosquitoes from three other states of India also revealed that this virus is abundantly present in these mosquitoes. Furthermore, presence of the virus at all developmental stages of the mosquito indicated vertical transmission of PCLV. Since PCLV is insect specific virus and there is no intermediate vertebrate host that can amplify the virus, it appears that transovarial transmission is the only way to maintain the virus.

Importantly, comparative sequence analysis showed >99% homology between Indian isolates as well as in isolates of other countries, suggesting high genetic stability of the virus. Presence of PCLV in natural populations of *Ae. aegypti* without significant changes in viral genomes suggests its possible endosymbiotic or commensal relationship with the mosquito without harming the fitness of its host.

Detection of PCLV in *Ae. aegypti* mosquitoes has already been reported from Brazil, Grenada, China, Thailand, USA etc [13,14,24,25,34]. However, the implication of PCLV in *Ae. aegypti* mosquitoes in the inhibition of pathogenic viruses such as dengue or chikungunya is not clearly known. Entomological investigations during dengue/chikungunya outbreaks in different parts of India have shown transmission by *Ae. aegypti* mosquitoes despite harboring high density of PCLV suggestive of its negative role in blocking the virus transmission (NIV unpublished data). This is in agreement with *in vitro* studies by Fredericks et al [5] where they have shown persistently infected cells harboring PCLV, did not show any impact on culturing and growth of other arboviruses. Being a member of the family *Peribunyaviridae*, which comprises several viruses of public health importance; the role of PCLV in interfering with replication of bunyaviruses is important and warrants investigation. Presence of ISVs belonging to family *Flaviviridae*, *viz.*, Bagaza virus, Palm Creek virus etc., have shown inhibition of JEV, WNV and Murray Valley encephalitis virus in their respective vectors [9,10]. Experimental studies have shown the phenomenon of superinfection exclusion where cultures/mosquitoes

infected with one flavivirus, reduced replication of another flavivirus upon subsequent infection [35].

Not much is still known about the role played by PCLV and other viruses in *Aedes aegypti* mosquitoes in the replication and transmission of human pathogenic viruses, *viz.*, dengue, chikungunya, Zika or Yellow fever. Further research into these aspects may come up with the potential of using the mosquito specific viruses in managing/impacting *Ae. aegypti* borne pathogenic viruses especially in the absence of prophylactics or therapeutics.

## Supporting information

**S1 Fig. Detection of PCLV S, M and L segments from Pune adult female mosquito pools using RT-PCR.** Products were analysed on 2% agarose gel, lanes- L1: Complete S segment, L2: Partial M segment, L3: Partial L segment, L4: 100 bp DNA ladder, L5: Negative control.
(TIFF)

**S2 Fig. Detection of PCLV S segment from mosquito samples from different locations from India using RT-PCR.** Products were analyzed on 2% agarose gel, lanes- L1: 100 bp DNA ladder (Invitrogen), L2: Assam (AS) adult mosquito pool, L3: Pune, Maharashtra (MH) adult mosquito pool, L4: Karnataka (KA) adult mosquito pool, L5: Tamil Nadu (TN) mosquito eggs, L6: Tamil Nadu (TN) mosquito larvae, L7: Tamil Nadu (TN) mosquito pupae, L8: Tamil Nadu (TN) mosquito adults, L9: 100 bp DNA ladder.
(TIF)

**S1 Table. Reads of representative sequencing runs after basecalling, adaptor and primer trimming.**
(DOCX)

**S2 Table. PCLV genome coverage in samples from different locations in Pune.**
(DOCX)

**S3 Table. GenBank accession numbers for 'S' segment sequences and sample details.**
(DOCX)

**S1 File.**
(PDF)

## Acknowledgments

The authors wish to thank Professor Priya Abraham, the Director, ICMR- NIV, for her keen interest in the project and for the constant support. The authors also thank Dr Ashwani Kumar and Dr R Paramsivan from ICMR-Vector Control Research Centre, Puducherry and Dr Siraj Khan, Regional Medical Research Centre Dibrugarh, Assam for providing mosquito eggs for the study.

## Author Contributions

**Conceptualization:** Kavita Lole, Anakkathil B. Sudeep, Sarah Cherian.

**Data curation:** Ashwini Ramdasi, Sucheta Patil, Shivani Thakar.

**Formal analysis:** Ashwini Ramdasi, Amol Nath.

**Funding acquisition:** Kavita Lole, Anakkathil B. Sudeep, Sarah Cherian.

**Investigation:** Amol Nath, Onkar Ghuge, Abhranil Gangopadhayya.

**Methodology:** Ashwini Ramdasi, Shivani Thakar, Amol Nath, Onkar Ghuge, Abhranil
Gangopadhayya.

**Project administration:** Sarah Cherian.

**Software:** Sucheta Patil, Onkar Ghuge, Abhranil Gangopadhayya.

**Validation:** Sucheta Patil.

**Writing – original draft:** Anakkathil B. Sudeep.

**Writing – review & editing:** Kavita Lole, Sarah Cherian.

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
