## [Decision Letter · Decision Letter 0]

26 Jul 2022

PONE-D-22-17294Abundance of Phasi-Charoen-like virus in Aedes aegypti mosquito populations in different states of IndiaPLOS ONE

Dear Dr. Sudeep,

Thank you for submitting your manuscript to PLOS ONE. After careful consideration, we feel that it has merit but does not fully meet PLOS ONE’s publication criteria as it currently stands. Therefore, we invite you to submit a revised version of the manuscript that addresses the points raised during the review process.

Authors must modify the discussion section regarding the IVS and other viruses also found. Please reorganize the tables and figures as suggested by one of the reviewers.

We look forward to receiving your revised manuscript.

Kind regards,

Humberto Lanz-Mendoza

Academic Editor

PLOS ONE

Journal Requirements:

3. PLOS requires an ORCID iD for the corresponding author in Editorial Manager on papers submitted after December 6th, 2016. Please ensure that you have an ORCID iD and that it is validated in Editorial Manager. To do this, go to ‘Update my Information’ (in the upper left-hand corner of the main menu), and click on the Fetch/Validate link next to the ORCID field. This will take you to the ORCID site and allow you to create a new iD or authenticate a pre-existing iD in Editorial Manager. Please see the following video for instructions on linking an ORCID iD to your Editorial Manager account: https://www.youtube.com/watch?v=_xcclfuvtxQ.

4. Please ensure that you refer to Figure 2 in your text as, if accepted, production will need this reference to link the reader to the figure.

Additional Editor Comments:

Authors must modify the discussion section regarding the IVS and other viruses also found. Please reorganize the tables and figures as suggested by one of the reviewers.

Reviewers' comments:

Reviewer's Responses to Questions

**Comments to the Author**

1. Is the manuscript technically sound, and do the data support the conclusions?

Reviewer #1: Yes

Reviewer #2: Partly

2. Has the statistical analysis been performed appropriately and rigorously? 

Reviewer #1: Yes

Reviewer #2: Yes

3. Have the authors made all data underlying the findings in their manuscript fully available?

Reviewer #1: Yes

Reviewer #2: Yes

4. Is the manuscript presented in an intelligible fashion and written in standard English?

Reviewer #1: Yes

Reviewer #2: Yes

5. Review Comments to the Author

Reviewer #1: Great manuscript! It opens the potential for future vector-based vaccines to kill dengue viruses inside the Aedes aegypti gut and salivary glands. These findings also may help to understand false positive ZIKV when PCR testing has been conducted in Culex, a non vector, in some locations of Brazil and Mexico. We invite these authors to pursue additional research efforts with rickettsial bacteria and related insect-living microorganisms

Reviewer #2: In the paper titled Abundance of Phasi-Charoen-like virus in Aedes aegypti mosquito populations in different states of India, by Kavita Lole and collaborators, the authors found the predominance of Phasi Charoen-like virus in Aedes aegypti mosquitoes field collected as well as, the presence of transovarial transmission verified in eggs, larvae, pupae, and male adults.

The data are interesting in the field of study and contribute to elucidating the virome of field mosquitoes.

However, as in other works, this article does not abound with evidence on the role that ISVs could have on infection with medically important viruses.

The discussion could focus on other viruses also found, on transovarian transmission, and the lack of evidence of viruses of medical importance in the samples.

There are tables that could be in supplementary material, and a map with the collected points is suggested. The presentation of the tables in the text should be in order and the legend figures should be more explanatory.

Introduction.

Line 71-73. The authors mention “These studies have shown inhibition/ reduced replication of Japanese encephalitis, Murray Valley encephalitis and West Nile viruses in respective vector mosquitoes that were experimentally infected with ISVs”. However in reference 10 in effect, they show inhibition or reduced replication but they observed it in cell culture, the authors should specify this point.

Line 75. The original reference is in Virus Genes volume55, pages 127–137 (2019)

Materials and Methods.

Line 86. Define which keys were used.

Line 89. The larvae reared until adults.

Line 98. Region specific primers (Table 1?)

Line 113. Rnase I ?

Line 118, Were these mosquitoes collected? Or are they the mosquitoes of the laboratory colony?

Line 144; Table 1. The author should reference where the random anchored primers were taken, (maybe reference 11).

Line 197. Space between table 1 and [11].

Line 206. These mosquitoes are representatives of the colony, right?

Line 210. In bold

Line 213. Mistake at the end f the sentence.

Results. Should the author clarify that the mosquito lysate spiked with cell culture with different viruses was done in order to have the best conditions to carry out the metagenomic virome analysis of field mosquitoes. The way it is currently written leaves some confusion, it should be clearer for the reader.

Why the author optimized the female mosquito lysate with cell culture virus. Wouldn't it have been better to have food mosquitoes (infected) with viruses as a better control?

Line 219. Mosquito of the laboratory colony.

Line 224. P5, P10, P12, and IDT-K

Line 243. I suggest that tables 2, 5, 6 could be presented as supplementary material.

Line 271. I can't see the three amplification products in figure 1a. All amplified products have the same size. The legend of the figure is not descriptive with what is shown

Line 298-300. Again the question. These mosquitoes are of the laboratory colony?

Line 304. This table is not necessary, this could resume in the text.

Discussion.

Line 376. the role of PCLV… I suggest the authors include a reference and discuss more about this point.

Most of the discussion is about the potential inhibitor effect of ISV against vector-borne disease, however, there is no evidence of this inhibitory effect as the authors mention in lines 367-369. Therefore I would like to see more data about this point in this work.

The most relevant data are in table 4. I suggest to the authors focus on the others virus found in field mosquitoes and the implications or consequences of these.

Interestingly in all field mosquito samples, the authors found no medically important flavivirus, so it would be nice if they could include it in the discussion if this will be because there are no current outbreaks or the possible presence of ISVs in particular PCLV.

References:

Line 425. erase space between 1371/journal to active correctly the hyperlink

6. PLOS authors have the option to publish the peer review history of their article (what does this mean?). If published, this will include your full peer review and any attached files.

Reviewer #1: No

Reviewer #2: No

---

## [Author Response · Author response to Decision Letter 0]

1 Sep 2022

PONE-D-22-17294

Abundance of Phasi-Charoen-like virus in Aedes aegypti mosquito populations in different states of India

PLOS ONE

Dear Dr. Sudeep,

Thank you for submitting your manuscript to PLOS ONE. After careful consideration, we feel that it has merit but does not fully meet PLOS ONE’s publication criteria as it currently stands. Therefore, we invite you to submit a revised version of the manuscript that addresses the points raised during the review process.

Authors must modify the discussion section regarding the IVS and other viruses also found. Please reorganize the tables and figures as suggested by one of the reviewers.

We look forward to receiving your revised manuscript.

Kind regards,

Humberto Lanz-Mendoza

Academic Editor

PLOS ONE

Journal Requirements:

2. PLOS ONE now requires that authors provide the original uncropped and unadjusted images underlying all blot or gel results reported in a submission’s figures or Supporting Information files

3. PLOS requires an ORCID iD for the corresponding author in Editorial Manager on papers submitted after December 6th, 2016. 

4. Please ensure that you refer to Figure 2 in your text as, if accepted, production will need this reference to link the reader to the figure.

Additional Editor Comments:

1) Authors must modify the discussion section regarding the IVS and other viruses also found. Please reorganize the tables and figures as suggested by one of the reviewers. 4. Is the manuscript presented in an intelligible fashion and written in standard English?

Reviewer #1: Yes

Reviewer #2: Yes

5. Review Comments to the Author

Reviewer #1: Great manuscript! It opens the potential for future vector-based vaccines to kill dengue viruses inside the Aedes aegypti gut and salivary glands. These findings also may help to understand false positive ZIKV when PCR testing has been conducted in Culex, a non vector, in some locations of Brazil and Mexico. We invite these authors to pursue additional research efforts with rickettsial bacteria and related insect-living microorganisms

Reviewer #2: In the paper titled Abundance of Phasi-Charoen-like virus in Aedes aegypti mosquito populations in different states of India, by Kavita Lole and collaborators, the authors found the predominance of Phasi Charoen-like virus in Aedes aegypti mosquitoes field collected as well as, the presence of transovarial transmission verified in eggs, larvae, pupae, and male adults.

The data are interesting in the field of study and contribute to elucidating the virome of field mosquitoes.

However, as in other works, this article does not abound with evidence on the role that ISVs could have on infection with medically important viruses.

The discussion could focus on other viruses also found, on transovarian transmission, and the lack of evidence of viruses of medical importance in the samples.

There are tables that could be in supplementary material, and a map with the collected points is suggested. The presentation of the tables in the text should be in your order and the legend figures should be more explanatory.

Response to reviewer's comments

Answer: Tables 2, 5 and 6 are shifted in the supplementary material as suggested. A map with sample collection points is added as Figure 1. The figure legends are now modified.

Introduction.

Line 71-73. The authors mention “These studies have shown inhibition/ reduced replication of Japanese encephalitis, Murray Valley encephalitis and West Nile viruses in respective vector mosquitoes that were experimentally infected with ISVs”. However in reference 10 in effect, they show inhibition or reduced replication but they observed it in cell culture, the authors should specify this point.

Answer: This has been now mentioned in the text (lines 70-74).

Line 75. The original reference is in Virus Genes volume55, pages 127–137 (2019)

Answer: The original reference ‘Öhlund, P., Lundén, H. & Blomström, AL. Insect-specific virus evolution and potential effects on vector competence. Virus Genes 55, 127–137 (2019). https://doi.org/10.1007/s11262-018-01629-9’ has been added in the list (ref 11)

Materials and Methods.

Line 86. Define which keys were used.

Answer: The keys described by Barraud PJ is included in the revised version (Ref. No. 15)

Line 89. The larvae reared until adults.

Answer: The line has been changed as suggested.

Line 98. Region specific primers (Table 1?)

Answer: Table 1 is now mentioned in the text as suggested.

Line 113. Rnase I ?

Answer: We are extremely sorry for this mistake. It is RNase A. This has been corrected in the text.

Line 118, Were these mosquitoes collected? Or are they the mosquitoes of the laboratory colony?

Answer: These were taken from a laboratory colony. This is now mentioned in the text.

Line 144; Table 1. The author should reference where the random anchored primers were taken, (maybe reference 11)

Answer: We are sorry for that. The references for primers are now added in the table 1. These references were already added in the list.

Line 197. Space between table 1 and [11].

Answer: The space has been added. Reference 11 is now 13 in the revised version.

Line 206. These mosquitoes are representatives of the colony, right?

Answer: These were either field-caught adult mosquitoes or larvae reared to adults. They were not from the colony.

Line 210. In bold

Answer: The subtitle, ‘Phylogentic analysis’ is now made bold.

Line 213. Mistake at the end f the sentence.

Answer: We are sorry for that. The sentence is corrected now.

Results. 

Should the author clarify that the mosquito lysate spiked with cell culture with different viruses was done in order to have the best conditions to carry out the metagenomic virome analysis of field mosquitoes. The way it is currently written leaves some confusion, it should be clearer for the reader.

Answer: We have added a line in the text (lines 223-225). Thank you for this suggestion.

Why the author optimized the female mosquito lysate with cell culture virus. Wouldn't it have been better to have food mosquitoes (infected) with viruses as a better control?

Answer: Yes, that would have been an ideal control. We used spiking approach so that we can try a range of virus concentrations for spiking during optimization experiments. 

Line 219. Mosquito of the laboratory colony.

Answer: We have now mentioned that,’ Lysates prepared from the laboratory colony of mosquitoes were spiked with…………as suggested (line 221).

Line 224. P5, P10, P12, and IDT-K

Answer: This has been mentioned in the text now (line 226).

Line 243. I suggest that tables 2, 5, 6 could be presented as supplementary material.

Answer: The tables 2, 5 and 6 are now shifted in the supplementary material. The same is also mentioned in the text.

Line 271. I can't see the three amplification products in figure 1a. All amplified products have the same size. The legend of the figure is not descriptive with what is shown

Answer: We are extremely sorry for this inadvertent mistake. We did not include the gel picture representing S, M and L segment amplicons from Pune mosquitoes. The picture given as figure 1a was S region amplification from different states of India and different developmental stages of mosquito. We have now added a new gel picture for S, M and L segment amplification as figure 1a. The earlier gel picture in now figure 1b. 

Line 298-300. Again the question. These mosquitoes are of the laboratory colony?

Answer: Yes, it was the laboratory colony. We obtained Ae. aegypti eggs from ICMR-Vector control Research Centre, Madurai, Tamil Nadu. These eggs were used to establish a colony to obtain larvae, pupae, adults and eggs. We have now modified the lines as, ‘To determine whether transovarial transmission of PCLV occurs in Ae. aegypti mosquitoes, a colony was established from Ae. aegypti eggs obtained from Madurai (Tamil Nadu state), larvae, pupae, adult and eggs from the colony were processed for S gene amplification.

Line 304. This table is not necessary, this could resume in the text.

Answer: Table 8 is deleted and the results are given in the text (lines 307-309) as suggested.

Discussion.

Line 376. the role of PCLV… I suggest the authors include a reference and discuss more about this point.

Most of the discussion is about the potential inhibitor effect of ISV against vector-borne disease, however, there is no evidence of this inhibitory effect as the authors mention in lines 367-369. Therefore I would like to see more data about this point in this work.

Answer: We have initiated experiments to isolate PCLV in C6/36 cells and hope to generate information on influence of PCLV on other arboviruses. As mentioned in the discussion (lines 409-412) we have seen no effect of PCLV on DENV and CHIKV transmission. We investigated one chikungunya outbreak recently (May 2022). We collected Ae. aegypti larvae obtained from containers from a patient’s house, reared to adults and processed for metagenomic analysis and obtained similar results (not included in this manuscript). These findings indirectly show that there is no influence of PCLV on CHIKV dissemination.

The most relevant data are in table 4. I suggest to the authors focus on the others virus found in field mosquitoes and the implications or consequences of these.

Answer: We have added more discussion on the other viruses detected in field caught mosquitoes (lines 385-406).

Interestingly in all field mosquito samples, the authors found no medically important flavivirus, so it would be nice if they could include it in the discussion if this will be because there are no current outbreaks or the possible presence of ISVs in particular PCLV.

Answer: This is now mentioned in the text (lines 406-408)

References:

Line 425. erase space between 1371/journal to active correctly the hyperlink

Answer: The space is removed

---

## [Decision Letter · Decision Letter 1]

25 Oct 2022

Abundance of Phasi-Charoen-like virus in Aedes aegypti mosquito populations in different states of India

PONE-D-22-17294R1

Dear Dr. Sudeep,

We’re pleased to inform you that your manuscript has been judged scientifically suitable for publication and will be formally accepted for publication once it meets all outstanding technical requirements.

Kind regards,

Humberto Lanz-Mendoza

Academic Editor

PLOS ONE

Additional Editor Comments (optional):

Reviewers' comments:

Reviewer's Responses to Questions

**Comments to the Author**

1. If the authors have adequately addressed your comments raised in a previous round of review and you feel that this manuscript is now acceptable for publication, you may indicate that here to bypass the “Comments to the Author” section, enter your conflict of interest statement in the “Confidential to Editor” section, and submit your "Accept" recommendation.

Reviewer #1: All comments have been addressed

Reviewer #2: All comments have been addressed

2. Is the manuscript technically sound, and do the data support the conclusions?

Reviewer #1: Yes

Reviewer #2: Yes

3. Has the statistical analysis been performed appropriately and rigorously? 

Reviewer #1: Yes

Reviewer #2: N/A

4. Have the authors made all data underlying the findings in their manuscript fully available?

Reviewer #1: Yes

Reviewer #2: Yes

5. Is the manuscript presented in an intelligible fashion and written in standard English?

Reviewer #1: Yes

Reviewer #2: Yes

6. Review Comments to the Author

Reviewer #1: Authors have considerede my comments. I really believe this is a very interesting manuscript. Research work related to arboviruses need to integrate these ISV results with PCR results and correlate local human diseases.

Reviewer #2: The authors have adequately addressed my comments raised in a previous round of review, this manuscript is now acceptable for publication.

7. PLOS authors have the option to publish the peer review history of their article (what does this mean?). If published, this will include your full peer review and any attached files.

Reviewer #1: **Yes: **Ildefonso Fernandez-Salas

Reviewer #2: **Yes: **Jorge Cime

---

## [Editor Report · Acceptance letter]

14 Nov 2022

PONE-D-22-17294R1 

Abundance of Phasi-Charoen-like virus in *Aedes aegypti* mosquito populations in different states of India 

Dear Dr. Sudeep:

I'm pleased to inform you that your manuscript has been deemed suitable for publication in PLOS ONE. Congratulations! Your manuscript is now with our production department. 

Kind regards, 

on behalf of

Dr. Humberto Lanz-Mendoza 

Academic Editor

PLOS ONE